# Magnetic Vortex and Hyperthermia Suppression in Multigrain Iron Oxide Nanorings

**Raja Das** [1,2,*], **Chiran Witanachchi** [3], **Zohreh Nemati** [3], **Vijaysankar Kalappattil** [3], **Irati Rodrigo** [4], **José Ángel García** [4,5], **Eneko Garaio** [5,6], **Javier Alonso** [3,7], **Vu Dinh Lam** [8], **Anh-Tuan Le** [9], **Manh-Huong Phan** [3] and **Hariharan Srikanth** [3,*]

1   Faculty of Materials Science and Engineering and Phenikaa Institute for Advanced Study (PIAS), Phenikaa University, Hanoi 10000, Vietnam

2   Phenikaa Research and Technology Institute (PRATI), A&A Green Phoenix Group, 167 Hoang Ngan, Hanoi 10000, Vietnam

3   Department of Physics, University of South Florida, Tampa, Tampa, FL 33620, USA; chiranw@mail.usf.edu (C.W.); znematip@umn.edu (Z.N.); vijaysankar8819@gmail.com (V.K.); jalonsomasa@gmail.com (J.A.); phanm@usf.edu (M.-H.P.)

4   BCMaterials, Basque Center for Materials, Applications and Nanostructures, UPV/EHU Science Park, 48940 Leioa, Spain; iratirodrigo@gmail.com (I.R.); joseangel.garcia@ehu.eus (J.Á.G.)

5   Depto. de Física Aplicada II, Universidad del País Vasco (UPV/EHU), 48940 Leioa, Spain; egarayo@gmail.com

6   Departamiento de Ciencias, Universidad Pública de Navarra (UPN), 31006 Pamplona, Spain

7   Depto CITIMAC, Universidad de Cantabria, 39005 Santander, Spain

8   Gradduate University of Science and Technology (GUST), Vietnam Academy of Science and Technology, Hoang Quoc Viet 18, Ha Noi, Vietnam; lamvd@ims.vast.ac.vn

9   Phenikaa University Nano Institute (PHENA), Phenikaa University, Hanoi 12116, Vietnam; tuan.leanh@phenikaa-uni.edu.vn

*   Correspondence: raja@phenikaa-uni.edu.vn (R.D.); sharihar@usf.edu (H.S.)

**Abstract:** Single-crystal iron oxide nanorings have been proposed as a promising candidate for magnetic hyperthermia application because of their unique shape-induced vortex-domain structure, which supports good colloidal stability and enhanced magnetic properties. However, the synthesis of single crystalline iron oxide has proven to be challenging. In this article, we showed that chemically synthesized multigrain magnetite nanorings disfavor a shape-induced magnetic vortex-domain structure. Our results indicate that the multigrain $Fe_3O_4$ nanorings with an average outer diameter of ~110 nm and an inner to outer diameter ratio of ~0.5 do not show a shape-induced vortex-domain structure, which was observed in the single-crystal $Fe_3O_4$ nanorings of similar dimensions. At 300 Ks, multigrain magnetite nanorings showed an effective anisotropy field of 440 Oe, which can be attributed to its high surface area and intraparticle interaction. Both calorimetric and AC loop measurements showed a moderate inductive heating efficiency of multigrain magnetite nanorings of ~300 W/g at 800 Oe. Our results shed light on the magnetic ground states of chemically synthesized multigrain $Fe_3O_4$ nanorings.

**Keywords:** multigrain; nanorings; magnetic vortex-domain; hyperthermia

## 1. Introduction

Spinel ferrite nanoparticles, $MFe_2O_4$ ($M = 3 - d$ transition metal) have attracted considerable attention in the past few decades due to their potential biomedical applications, for example in targeted drug delivery, diagnostics, and magnetic separation [1–13]. Spinel ferrite nanoparticles have been

mostly studied because of their unique size- and shape-dependent tunable magnetic properties [14–18]. For biomedical applications, iron oxide (magnetite or maghemite) has been considered as the most attractive material because of its intrinsic biocompatibility and chemical stability [19–21]. Large surface to volume ratio, size- and shape-tunable magnetic properties, and high biocompatibility make iron oxide nanoparticles appropriate nanocarriers for magnetic resonance imaging, magnetic hyperthermia, and targeted drug delivery [22–30].

Magnetic hyperthermia has emerged as a promising alternative to the currently used cancer treatment methods, such as chemotherapy and radiotherapy, which have severe side effects [22–30]. When iron oxide nanoparticles are exposed to an external alternating current (AC) magnetic field, they convert the electromagnetic energy into thermal energy [22–30]. Magnetic nanoparticles act as nanometric heating centers that one can use to target a specific tumor area and deliver toxic doses of heat to the tumor area without affecting the neighboring healthy tissues, thus making magnetic hyperthermia a less aggressive and more effective method of targeted cancer treatment. Magnetic hyperthermia using superparamagnetic iron oxide nanoparticles is currently being implemented in combination with chemotherapy or radiotherapy, allowing a reduction of the chemotherapeutic drug dose needed, and yielding several encouraging outcomes in cancer treatment [31–33]. However, the low heating efficiency of these nanoparticles presents serious concerns about the dose of nanoparticles necessary for an effective cancer treatment [34]. The heating efficiency (estimated by specific absorption rate, SAR) of magnetic nanoparticles depends on their intrinsic properties, including the saturation magnetization and magnetic anisotropy [27–30]. Size reduction has been reported to alter the saturation magnetization and magnetic anisotropy due to poor crystallinity, which could negatively impact the heating efficiency of the nanoparticles [27–30]. Additionally, the SAR value of the nanoparticles highly depends on the colloidal stability or inter-particle interaction of the nanoparticles. To overcome these limits, different strategies have been proposed, such as tuning the shape and size of these nanoparticles. In this regard, single-crystalline iron oxide nanorods [27] and nanotubes [28] have recently been synthesized, which have shown excellent magnetic hyperthermia properties. Single-crystalline iron oxide vortex nanorings with high saturation magnetization and negligible remanence and coercivity have also been reported to show excellent magnetic hyperthermia response [35,36]. However, control over the synthesis of these single-crystalline iron oxide vortex nanorings is a challenging task, raising serious doubt about their practical application [35–41]. Electron beam lithography and chemical methods have been employed to fabricate single-crystalline vortex nanorings, but a large-scale controlled synthesis of high-quality, defection-free single-crystalline iron oxide nanorings has not been achieved yet [35,36]. Multigrain iron oxide nanorings are easy to form chemically. A full understanding of the magnetic and hyperthermia responses of these multigrain iron oxide nanorings will provide important insights into the magnetic vortex nature and the large SAR of the single-crystalline vortex iron oxide nanorings.

In this study, we have chemically synthesized multigrain magnetite nanorings with an average outer diameter ~120 nm and an inner to outer diameter ratio of ~0.3–0.5. At room temperature, the multigrain $Fe_3O_4$ nanorings showed typical characteristics of a ferromagnetic system. Both calorimetric and AC loop measurements yielded moderate SAR values for all magnetite nanoring samples investigated. Our study pinpoints that it is the formation of the multigrain structure that disfavors the formation of a magnetic vortex and causes the reduction of the heating efficiency in magnetite nanorings.

## 2. Experimental Section

### 2.1. Synthesis of $Fe_3O_4$ Nanorings

$Fe_3O_4$ nanorings were synthesized using a previously reported two-step method [35,36,42–44]. First, $\alpha$-$Fe_2O_3$ nanorings were synthesized using hydrothermal reaction of $FeCl_3$ with $NaH_2PO_4$ and $Na_2SO_4$. Then, the $\alpha$-$Fe_2O_3$ nanorings were reduced to form $Fe_3O_4$ nanorings. In a typical reaction, specific amounts of $FeCl_3 \cdot 6H_2O$ (0.27 g), $NaH_2PO_4 \cdot 2H_2O$ (0.014 g), and $Na_2SO_4$ (0.0195 g) are added to



35 mL of water and stirred at room temperature for 30 min. Afterwards, the reaction mixture is poured into a Teflon-lined stainless steel autoclave and heated at 220 °C for various amounts of time. After letting the autoclave cool down naturally to room temperature, the red colored precipitate is washed three times using a mixture of ethanol and water. The reduction of as-synthesized dried $\alpha$-$Fe_2O_3$ nanoring powder was done in the presence of hydrogen/argon (7% hydrogen) at 300 °C for 5 h to produce $Fe_3O_4$ nanorings.

## 2.2. Characterization

The crystal structure of the nanoparticles was characterized using a Bruker AXS D8 X-ray diffractometer (XRD). The morphology of the nanoparticles was characterized with a FEI Morgagni 268 transmission electron microscope (TEM) operating at 60 kV. Magnetic hysteresis loops and magnetization vs. temperature curves were recorded using a vibrating sample magnetometer (VSM) attachment for Quantum Design, Physical Property Measurement System (PPMS). Calorimetric hyperthermia measurements were performed using a 4.2-kW Ambrell Easyheat Li3542 system (glass vial of 16 × 50 mm), with (AC) fields in the range of 0–800 Oe at a frequency of 310 kHz. The dynamic hysteresis loops in the range of 0–400 Oe were measured using a homemade AC magnetometer setup at a frequency of 302 kHz [45].

## 3. Results and Discussion

The phase purity and crystallinity of the nanorings were examined by XRD. Figure 1 shows the representative XRD spectra of $\alpha$-$Fe_2O_3$ after 5 h of reaction and reduced $Fe_3O_4$ nanorings. Both the as-synthesized and annealed samples display sharp peaks, which can be indexed to $\alpha$-$Fe_2O_3$ and $Fe_3O_4$, respectively. The sharp XRD peaks, evidence of high crystallinity, provide an advantage to the $Fe_3O_4$ nanorings, as this ensures improved magnetic property, and thus better inductive heating efficiency. The full width at half maxima (FWHM) of all the diffraction peaks of $Fe_3O_4$ nanorings (5 h of reaction) are relatively large, suggesting a small crystallite size of the $Fe_3O_4$. The average crystallite size of $Fe_3O_4$ (5 h of reaction) as calculated using the Debye–Scherrer formula is 15 nm.

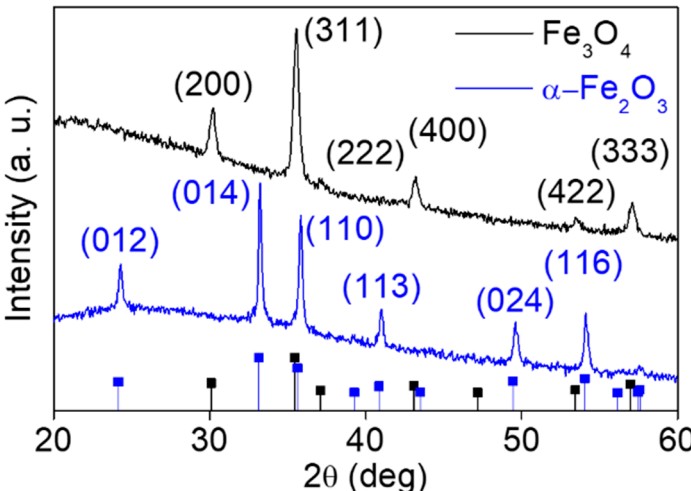

**Figure 1.** X-ray diffraction (XRD) pattern of as-synthesized ($\alpha$-$Fe_2O_3$) and reduced ($Fe_3O_4$) nanorings after 5 h of hydrothermal reaction. The lower patterns with blue and black squares are the JCPDS data for bulk $\alpha$-$Fe_2O_3$ and $Fe_3O_4$, respectively [21,27,28].

To further study the microstructure and morphology of $Fe_3O_4$ nanorings, TEM imaging of the annealed samples was performed. We have shown earlier that the morphology and size of the $\alpha$-$Fe_2O_3$ nanoparticles does not change with annealing [28]. TEM images in Figure 2 showed that the annealed $Fe_3O_4$ nanoparticles obtained after hydrothermal reaction at 220 °C exhibit ring morphology and their

size is relatively polydispersed. The annealed sample obtained after 5 h of hydrothermal reaction (Figure 2a,b) yielded nanorings with an average outer diameter of ~110 nm and an inner to outer diameter ratio of ~0.55. As can be seen from the TEM image in the inset, the $Fe_3O_4$ nanorings seem to consist of many semispherical $Fe_3O_4$ nanoparticles of around 20 nm, which are self-assembled to form a multigrain nanoring. This supports the observation of broad XRD peaks on the $Fe_3O_4$ nanoring, which had a crystallite size of 15 nm. To understand the effect of hydrothermal reaction time on the evaluation of nanoring morphology and dimensions, we varied the reaction time while keeping other reaction parameters unperturbed. TEM images of the annealed multigrain $Fe_3O_4$ nanorings obtained after 8 and 12 h of hydrothermal reaction (Figure 2c,d) yielded particles with average outer diameter of ~110 and 105 nm, respectively, and an inner to outer diameter ratio of ~0.5. The average outer diameter and an inner to outer diameter ratio of all the nanorings can be seen in Table 1. The time-dependent studies of hydrothermal reaction suggest that the nanoring morphology and dimensions remain almost the same from 3 to 12 h of reaction. TEM images of the reduced $Fe_3O_4$ nanorings after 12 h of hydrothermal reaction (Figure 2d) indicated that prolonging the reaction time causes agglomeration and breaking of the multigrain nanorings. This was also observed in the case of nanotubes, where we showed breakdown of nanotubes after 24 h of reaction [28].

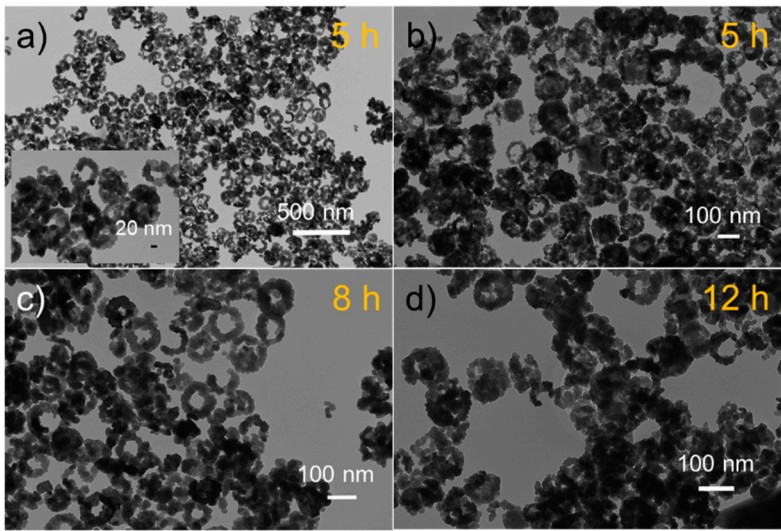

**Figure 2.** Representative bright-field TEM images of multigrain $Fe_3O_4$ nanorings after hydrothermal reaction time: (**a,b**) 5 h, (**c**) 8 h, (**d**) 12 h. Inset of (**a**) shows a magnified view of the nanorings.

**Table 1.** Crystallite size, dimensions, magnetic parameters including saturation ($M_S$) and coercive field ($H_C$), and specific absorption rate (SAR) at 300 K for $Fe_3O_4$ nanorings.

| Reaction Time (h) | Crystallite Size (nm) | Outer Diameter (nm) | Inner to Outer Diameter Ratio (β) | $M_S$ (emu/g) | $H_C$ (Oe) | SAR in Water at 800 Oe (W/g) |
|---|---|---|---|---|---|---|
| 3 | 15 | 100 | 0.5 | 52 | 150 | 155 |
| 5 | 15 | 110 | 0.55 | 54 | 130 | 262 |
| 8 | 17 | 110 | 0.5 | 55 | 140 | 342 |
| 12 | 18 | 105 | 0.5 | 57 | 155 | 186 |

Magnetic hysteresis loops measured at 300 K in multigrain $Fe_3O_4$ nanorings after different hydrothermal reaction times of 3, 5, 8, and 12 h are shown in Figure 3a. The saturation magnetization ($M_S$) and coercive field ($H_C$) values at 300 K of the $Fe_3O_4$ nanorings of different hydrothermal reaction times are summarized in Table 1. In Figure 3a, all the multigrain $Fe_3O_4$ nanoring samples show similar behavior at 300 K. Despite having small crystallite sizes, the presence of notable $H_C$ in all

the samples could be due to presence of defects, along with large intra- and inter-particles dipolar interaction of multigrain rings. The $H_C$ values of all the samples are almost the same. The shape of the loop and the presence of remanence and coercivity in the magnetic hysteresis (*M-H*) loops of multigrain $Fe_3O_4$ nanoring samples indicate a typical soft ferromagnetic character. It has been shown that the single-crystalline $Fe_3O_4$ nanoring samples with similar dimensions possess a shape-induced vortex-domain structure [35]. However, in the present case, we did not observe onion–vortex–onion or onion-to-onion structured ground states, despite the theoretical phase diagram suggesting the formation of a vortex-domain structure [37–41]. This result clearly indicates that the microstructure of rings play an important role in determining their magnetic ground state. Theoretically, it was predicted that even if the dimension of a nanorings falls in the vortex state in the ground state phase, it might get trapped in a metastable state because of defects or inter-particle distance [37–41]. The absence of a shape-induced vortex-domain structure in multigrain $Fe_3O_4$ nanorings could be related to these effects. The values of $M_S$ in the $Fe_3O_4$ nanoring samples increased gradually with the increase in reaction time. The $M_S$ values are in the range of 55 emu/g for all the $Fe_3O_4$ nanoring samples and are smaller than the theoretical magnetization value of bulk magnetite at room temperature (~92 emu/g). The small variation of the $M_S$ at 300 K with the change in hydrothermal reaction time can be attributed to the variation in size and crystalline quality of the nanorings. The zero-field-cooled (ZFC) magnetization vs. temperature curve for all the samples (not shown here) showed a sharp drop in magnetization below 120 K, which can be attributed to the Verwey transition linked with the metal insulator transition in $Fe_3O_4$ [27,28]. Due to poor crystalline quality, non-stoichiometric effect and defects, this transition—which is related to the crystal structure of $Fe_3O_4$—often gets suppressed in nanoparticles [46,47]. The occurrence of the Verwey transition in the $Fe_3O_4$ nanorings indicates the realization of high crystalline quality and single phase $Fe_3O_4$ nanostructures.

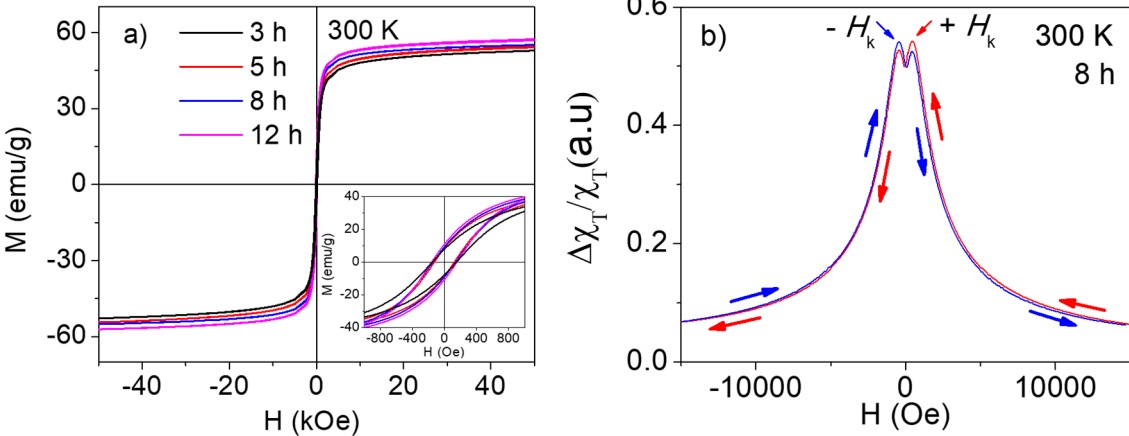

**Figure 3.** (**a**) Magnetic hysteresis loops of the multigrain $Fe_3O_4$ nanorings after different hydrothermal reaction times, measured at 300 K. (**b**) Representative bipolar transverse susceptibility (TS) curves taken at 300 K for multigrain $Fe_3O_4$ nanorings after 8 h of reaction. Inset of (**a**) shows the low-field region of the hysteresis loops.

The occurrence of the Verwey transition confirms the high crystalline quality of the multigrain $Fe_3O_4$ nanorings. The high coercivity, the ring shape, and strong inter- and intra-particle interactions of the samples indicate the high effective magnetic anisotropy of the multigrain $Fe_3O_4$ nanorings. The estimation of the effective magnetic anisotropy of the $Fe_3O_4$ nanorings was done using radio frequency (RF) transverse susceptibility (TS). We have established the TS method as a precise tool for investigating the anisotropic magnetic properties of a variety of systems, single crystals, thin films, and nanoparticles [48–51]. For a Stoner–Wohlfarth particle with its magnetic hard axis aligned parallel to the DC field, TS spectra should display peaks at the anisotropy fields ($\pm H_K$) and switching fields ($\pm H_S$) as the DC field is swept from positive to negative saturation [52].

Bipolar TS curves taken at 300 K for multigrain $Fe_3O_4$ nanorings after 8 h reaction are displayed in Figure 3b. We obtained no variation in the $H_K$ values for multigrain $Fe_3O_4$ nanorings after different hydrothermal reaction time of 3, 5, 8, and 12 h, so we have presented the results for multigrain $Fe_3O_4$ nanorings after 8 h of reaction. We have noted that for the $Fe_3O_4$ nanorings, the switching peak was merged with the anisotropic peak, resulting in a difference in the positive and negative $H_K$ values, along with a slight difference in the peak height (Figure 3b). Here, we have used a positive $H_K$ value. The presence of distinct $H_K$ peaks at 300 K in the TS spectra indicates high effective anisotropy in the $Fe_3O_4$ nanorings. In nanoparticles, the thermal fluctuations dominate anisotropy energy at room temperature; thus, sustaining magnetic anisotropy at room temperature in nanostructures is challenging [53]. The value of effective $H_K$ at 300 K for multigrain $Fe_3O_4$ nanorings after 8 h reaction was found to be 440 Oe, which is higher than the $H_C$ value of 140 Oe. The effective magnetic anisotropy, $H_K$, roughly corresponds to the coercive field in the non-interacting system of nanoparticles. The higher value of $H_K$ of multigrain $Fe_3O_4$ nanorings compared to $H_C$ could be due to the strong intra- and inter-nanoring interactions. Our recent reported works have revealed that the heating efficiency of magnetic nanostructures can be enhanced by increasing the saturation magnetization ($M_S$) or the shape and magnetic anisotropy ($H_K$) of the nanostructures [27,28].

To estimate the heating efficiency (SAR) of the multigrain $Fe_3O_4$ nanorings, the SAR values were calculated using both calorimetric and dynamic hysteresis loop measurements. For calorimetric measurements, the heating curves as a function of time were recorded for all the multigrain $Fe_3O_4$ nanorings at a concentration of 1 mg/mL, both in water and agar (random and aligned). By dispersing the nanorings in a 2% weight agar solution, we constrained the Brownian motion of the nanorings using the higher viscosity of 2% weight agar solution, and also fixed their orientation. A 2% weight agar solution also imitates the characteristics of the cell cytoplasm and extra cellular matrix, which is critical for in vivo testing of magnetic hyperthermia [54]. The heating efficiency of the nanorings was calculated using the initial slope method. In this method, the SAR of the nanoparticles can be calculated using the following equation:

$$\text{SAR} = \frac{\Delta T}{\Delta t} \frac{C_p}{]\varphi} \tag{1}$$

where $\frac{\Delta T}{\Delta t}$ represents the initial slope of the heating curves, $C_p$ is the specific heat capacity of the liquid medium (4.186 J/g °C for water), and $\varphi$ is the mass of magnetic material per unit mass of liquid.

In Figure 4a, we plotted the SAR values as a function of AC magnetic field strength when the nanorings samples were randomly dispersed in water at a concentration of 1 mg/mL. As can be seen from Figure 4a, with a 200 Oe applied field, the SAR values of all the samples were very small, which could be due to the fact that the effective $H_K$ values of the samples were close to 400 Oe. With the increase in the field above the effective $H_K$ values of the samples, the SAR values showed an enhancement, which further increased with increasing applied field. The heating efficiencies are almost the same for all the samples, except at 800 Oe. At the maximum applied field level, we can observe a noticeable increase in the SAR value for the 8 h sample. The slight variation of the SAR values at lower applied fields scan be attributed to the small variation in $M_S$, effective $H_K$, and the crystalline quality of the nanorings, but something different occurred for the 8 h sample with increasing applied field. The Néel and Brownian (physical rotation) processes contributed to the SAR values of the magnetic nanoparticles. To decouple the contribution from Néel and Brownian processes towards the SAR values, measurements were done by dispersing the nanorings in 2% weight agar. The high viscosity of the 2% weight agar confines the physical motion (Brownian relaxation) of the nanorings. The results showed that the heating efficiency of the randomly dispersed samples in agar was slightly reduced compared to the water-dispersed samples (Figure 4b). This showed that most of the contribution to the heating efficiency of these nanorings comes from the magnetic and not the physical rotation, which is important for clinical hyperthermia application, since physical rotation of the nanoparticles tends to be highly suppressed in the tumor environment [55]. The only exception to this trend was again the

8 h sample, which conserved nearly the same SAR values that were exhibited in water. In our earlier study on $Fe_3O_4$ nanorods and nanotubes, we observed a large enhancement in heating efficiency of the nanostructures when they were oriented in agar solution using a DC magnetic field compared to randomly oriented samples [27,28]. This was attributed to the improved magnetic anisotropy of the one-dimensional nanostructures when aligned. To study the effect of nanoring alignment on the heating efficiency, the samples were aligned in agar solution using a DC magnetic field. As can be seen from Figure 4c, the heating efficiency of all the samples at fields ≥ 400 Oe improved in aligned samples as compared to the randomly dispersed ones in agar. At 200 Oe (below the coercive field of the samples), the SAR values of all the samples remain almost constant in water and agar (random and aligned), because the applied field is lower than the anisotropy field, $H_K$. As the magnitude of the AC magnetic field was increased, the SAR values of the agar (aligned) samples increased compared to the water and agar (random) samples. An enhancement in SAR of up to 40% was witnessed in the 2% weight agar-aligned samples at the applied field range of 400–800 Oe compared to random agar samples. Moreover, all samples reached this time with very similar SAR values, including for the 8 h sample. This suggests that differences previously observed in the SAR values of the 8 h sample compared to the rest can be related to a better alignment or assembly of this sample, which allowed it to maximize the SAR value. The fact that the 8 h sample exhibits similar SAR values, independent of the medium and the conditions, may suggest that the magnetic nanorings in this sample have a tendency to assemble well [56], even in the absence of a magnetic field. This is probably due to a desirable combination of inter-particle or inter-nanoring interactions, morphology, or other factors. It is also worth mentioning that the 8 h sample exhibited increased SAR values as compared to the 12 h sample. If this was due to effect of the increased reaction time (i.e., increased grain size, saturation magnetization, etc.), one would expect to observe a larger SAR value for the 12 h sample, which did not happen in our case. A possible explanation is that as compared to the 8 h sample, the iron oxide nanorings, as we saw in TEM images (Figure 2), tend to degrade and agglomerate at 12 h, which would hinder their magnetic response at low fields, as indicated by the dynamic hysteresis loops in Figure 5. Nevertheless, further study is needed to fully understand this phenomenon, which is beyond the scope of the present paper. In addition, our results demonstrated that the SAR values of the multigrain $Fe_3O_4$ nanorings can be appreciably improved when nanorings align with the field direction.

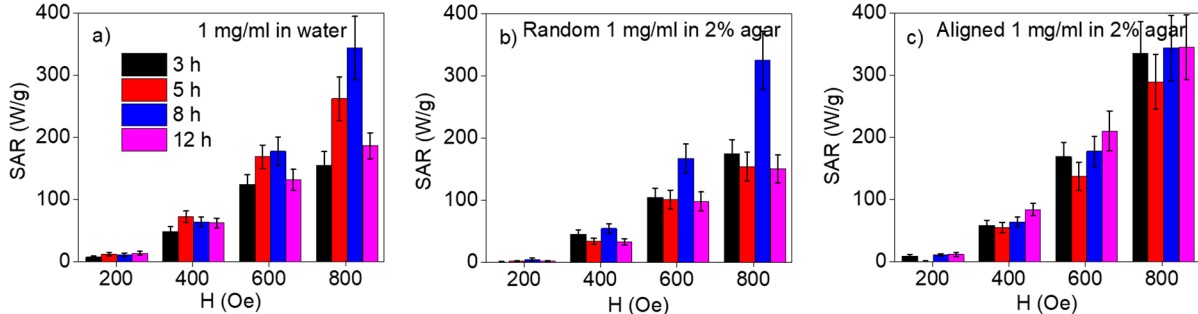

**Figure 4.** SAR vs. AC magnetic field plots for all the samples, dispersed in 1 mg/mL of (**a**) water, (**b**) 2% randomly suspended agar solution, and (**c**) pre-aligned to an external dc magnetic field.

From the calorimetric measurements, it was observed that the SAR values of the nanoring samples are relatively small at AC fields lower than the effective magnetic anisotropy. To understand the nature of heating in multigrain $Fe_3O_4$ nanorings at low fields, we performed the AC magnetometry measurements at fields < 400 Oe for the $Fe_3O_4$ nanorings (8 h and 12 h). Figure 5a–d show the SAR values and dynamic hysteresis loops in the range of 0-400 Oe at 302 Hz. The heating efficiency and SAR values of the nanoring samples were calculated from the hysteresis loop area (SAR = area × frequency) [28]. Figure 5c,d shows that at low fields the hysteresis loop resembles Rayleigh loops, and area of the loops are quite small. As the field was increased, the loop area showed a rapid increase [28].

From Figure 5a–d, it can be seen that the area of the hysteresis loop and the SAR values almost remained unchanged until reaching an AC field of ~200 Oe, and showed a steep increase when the AC field was increased above 200 Oe. It can be seen that for the 8 h sample, this increase is steeper and reaches higher SAR values, as was observed in calorimetric measurements. It should be noted that both the calorimetric and AC magnetometry measurements showed very similar SAR values. The AC magnetometry measurement results can be elucidated considering the ratio of the anisotropy field ($H_K$) to the applied AC field. Depending on the $H_K$ of the nanostructures, the dynamic hysteresis loop shows two regions. At low field levels, when $H < 0.5\,H_A$, since the power absorption is mainly caused by viscous losses in the system, a minor loop is obtained. Conversely, when the field $H > 0.5\,H_A$, the hysteresis losses dominate, maximum heat power is transferred to the nanostructures, and the hysteresis loop area increases. Below the applied field of 200 Oe, which is ~0.5 of $H_K$ of the nanorings (Figure 3b), the hysteresis loop area and the SAR values are negligible, while with the increase in the field, the hysteresis loop area and the SAR values showed a rapid increase. However, since the maximum AC applied field is still lower than $H_K$, the AC loop area is not maximized, and higher fields would be needed to saturate the AC loops [57]. Therefore, AC magnetometry measurements shed light on the low-field heating response of the nanorings that cannot be inferred from the calorimetric measurements.

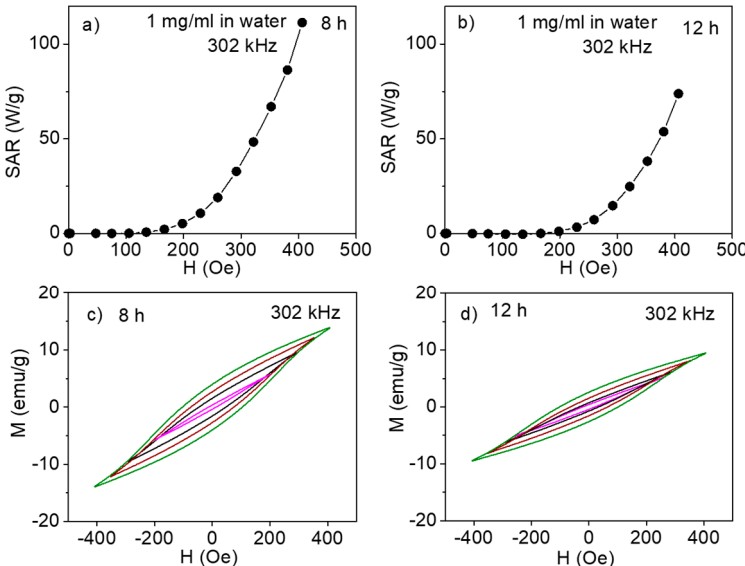

**Figure 5.** (**a**,**b**) SAR vs. AC magnetic field curves as obtained from (**c**,**d**) the corresponding dynamic hysteresis loops for multigrain Fe$_3$O$_4$ nanorings after 8 h and 12 h of reaction, respectively.

## 4. Conclusions

Highly crystalline, multigrain Fe$_3$O$_4$ nanorings were synthesized using a simple hydrothermal method. Systematic magnetic measurements indicated that the multigrain magnetite nanorings of an average outer diameter of ~110 nm and an inner to outer diameter ratio of ~0.5 showed typical characteristics of a ferromagnetic system at room temperature, unlike the predicated vortex-domain structure. Our experiments revealed that unlike the single-crystal Fe$_3$O$_4$ nanorings, the multigrain Fe$_3$O$_4$ nanorings did not show a shape-induced vortex-domain structure, despite having similar dimensions and morphology. The corroborative results of both calorimetric and AC loop measurements of the multigrain Fe$_3$O$_4$ nanorings showed moderate inductive heating efficiency. Interestingly, alignment of the nanorings with the applied field allowed them to maximize their SAR values. Our results demonstrated the possibility of using Fe$_3$O$_4$ nanorings for dual-purpose applications in localized magnetic hyperthermia therapy and controlled drug delivery because of their highly effective anisotropy and hollow morphology.

**Author Contributions:** Conceptualization, R.D. and M.-H.P.; methodology, R.D., C.W., and I.R.; investigation, R.D., C.W., Z.N., V.K., and I.R.; formal analysis, R.D., J.A., I.R., and M.-H.P.; resources, M.-H.P., H.S., J.Á.G., and E.G.; writing—original draft preparation, R.D. and M.-H.P.; writing—review and editing, R.D., C.W., Z.N., V.K., J.A., I.R., J.Á.G., E.G., V.D.L., A.-T.L., M.-H.P., and H.S.; supervision, R.D., M.-H.P., and H.S. All authors have read and agreed to the published version of the manuscript.

**Acknowledgments:** Research at the University of South Florida was supported by the U.S. Department of Energy, Office of Basic Energy Sciences, Division of Materials Sciences and Engineering, under Award No. DE-FG02-07ER46438. This research was also supported by Vietnam Ministry of Science and Technology through the national-level project ĐTĐLCN.17/19.

**Conflicts of Interest:** The authors declare no conflict of interest.

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
