# Peer review of "Magnetic Vortex and Hyperthermia Suppression in Multigrain Iron Oxide Nanorings"

_applsci, doi:10.3390/app10030787_

Round 1

Reviewer 1 Report

This manuscript presents a study on the preparation of multi-grain iron oxide nanorings and on their magnetic properties, in particular on their heat generation. The authors used a two-step process developed in the literature to create first hematite multi-grain nanorings, which have been subsequently reduced to magnetite. The morphology, magnetization and SAR of the nanorings have been measured, the latter both in solution, and in agar gels (both random and aligned). The results shows that the morphology strongly depends on the reaction time, and consequently their magnetic properties.

The manuscript is quite well written, and the study is well conceived. The results are interesting, especially in terms of synthesis of the nanorings. I believe that the authors should however deepen the discussion on the peculiar behavior of the sample synthesized for 8 h. It is not clear at all what the authors mean by saying that the 8h sample “This suggests that differences previously observed in the SAR values of the 8h sample compared to 247 the rest can be related to a better alignment of this sample with increasing AC field, which allowed it 248 to maximize the SAR value “. How can the sample align when exposed to an AC field when immersed in an Agar gel? Furthermore, how was the alignment obtained? Details are missing…

In summary, the manuscript is worth published, but some additional discussions and information are necessary.

Reviewer 2 Report

The manuscript is concise and well written. Experiments are adequately co-ordinated and discussed jointly. It is worth printing after authors address the following minor comments-suggestions.

In hyperthermia interpretation, authors should clarify

in Fig3b the switching field for the 8h (440 Oe) sample  seems to sustain in hyperthermia fields up to 400 Oe while for bigger ones AC field significantly increases the SAR. What is the physical origin for such a big difference in SAR between 8 h and its neighborhood (5h and 12h). In all other cases where the switching field is smaller? this should also happen at a different threshold? Please explain 

Also the agar samples' discussion is misleading. The 2% agar indeed attenuating Brownian motion, but not to full extent?

What happens at arrays that practically their performance converges (SAR values -> 300 W/g at 800 Oe) regardless of their reaction time contrary to as prepared samples.

Figure 5c, 5d Legend to identify the loops is missing.

Round 2

Reviewer 1 Report

The authors have addressed the comments well. Publication is not recommended.